# Optimizing Polymer Costs and Efficiency in Alkali–Polymer Oilfield Applications

**DOI:** 10.3390/polym14245508

**Published:** 2022-12-15

**Authors:** Rafael E. Hincapie, Ante Borovina, Torsten Clemens, Eugen Hoffmann, Muhammad Tahir, Leena Nurmi, Sirkku Hanski, Jonas Wegner, Alyssia Janczak

**Affiliations:** 1OMV Exploration & Production GmbH, Trabrennstrasse 6-8, 1020 Vienna, Austria; 2HOT Microfluidics GmbH, Trabrennstrasse 6-8, 1020 Vienna, Austria; 3Kemira OYJ, Trabrennstrasse 6-8, 1020 Vienna, Austria

**Keywords:** polymer flooding, alkali–polymer flooding, chemically enhanced oil recovery, incremental oil recovery

## Abstract

In this work, we present various evaluations that are key prior field applications. The workflow combines laboratory approaches to optimize the usage of polymers in combination with alkali to improve project economics. We show that the performance of AP floods can be optimized by making use of lower polymer viscosities during injection but increasing polymer viscosities in the reservoir owing to “aging” of the polymers at high pH. Furthermore, AP conditions enable the reduction of polymer retention in the reservoir, decreasing the utility factors (kg polymers injected/incremental bbl. produced). We used aged polymer solutions to mimic the conditions deep in the reservoir and compared the displacement efficiencies and the polymer adsorption of non-aged and aged polymer solutions. The aging experiments showed that polymer hydrolysis increases at high pH, leading to 60% higher viscosity in AP conditions. Micromodel experiments in two-layer chips depicted insights into the displacement, with reproducible recoveries of 80% in the high-permeability zone and 15% in the low-permeability zone. The adsorption for real rock using 8 TH RSB brine was measured to be approximately half of that in the case of Berea: 27 µg/g vs. 48 µg/g, respectively. The IFT values obtained for the AP lead to very low values, reaching 0.006 mN/m, while for the alkali, they reach only 0.44 mN/m. The two-phase experiments confirmed that lower-concentration polymer solutions aged in alkali show the same displacement efficiency as non-aged polymers with higher concentrations. Reducing the polymer concentration leads to a decrease in EqUF by 40%. If alkali–polymer is injected immediately without a prior polymer slug, then the economics are improved by 37% compared with the polymer case. Hence, significant cost savings can be realized capitalizing on the fast aging in the reservoir. Due to the low polymer retention in AP floods, fewer polymers are consumed than in conventional polymer floods, significantly decreasing the utility factor.

## 1. Introduction

The selection of a chemically enhanced oil recovery (cEOR) agent, such as a polymer or an alkali, is critical for the correct design of technology applications. Polymer flooding has been implemented in numerous fields [1,2], although its role in reducing residual oil saturation remains a discussion topic. Due to its high viscosity, a polymer alone cannot significantly increase the capillary number; however, it can contribute to oil recovery, as its mobility can be controlled [3]. Polymer technology has been proven to be very effective in the recovery process using mobility control with multiple field applications [4].

Similarly, field applications of using an alkali alone as an EOR agent resulted in a lower oil recovery of 1–6% [5]. Although a chemical reaction between an alkali and crude oil generates in situ saponification, alkali consumption and plugging remain significant challenges [6,7,8]. The synergy of alkali and polymer injection can surpass the reported limitations of injecting them separately. Schumi et al. [7] and Hincapie et al. [9] reported that alkali–polymer (AP) slugs offer a higher incremental oil contribution than when injected alone (alkali/polymer/water) or as separate slugs (alkali after polymer/polymer after alkali) [10].

Furthermore, laboratory results indicated that less alkali was consumed using an AP slug than using an alkali slug [11]. One possible reason is that polymer molecules cover the rock surface to reduce alkali adsorption. Additionally, alkali increases pH and makes the rock surface charge more negative. This chemical process can hinder polymer adsorption at the rock surface [10,12]. As a result, more alkali reacts with the crude oil polar compounds to generate in situ soaps and lower interfacial tension. Furthermore, polymers hold the injected aqueous-phase viscosity and improve the sweep efficiency.

Project economics play a pivotal role for EOR technology applications. For a highly complex case such as AP involving multiple chemical agents, the business case for the project may become unfavorable or hard to justify [13]. With a proper process design, good recoveries can be observed, and it is possible that economics can be improved [9].

This work focuses on evaluating the synergies of alkali and polymers for a successful application of the technology in the Matzen field in Austria. For the optimum utility factor, aging of AP slugs is performed to optimize the project economics. This study is a continuation of our previous research [13,14], with an emphasis on AP flooding in reservoir sand packs and reservoir conditions (live oil).

The paper is organized as follows: in the next section, the approach is covered. Then, the materials and methods are covered, followed by the results and the discussion.

## 2. Approach

To evaluate the alkali–polymer synergies on recovery and to improve the economics, the following steps were followed:Fluid/rock selection and characterization: one polymer and one alkali type were selected and characterized. Oil properties were also measured.Aging: aging experiments were carried out to evaluate the changes in the long-term polymer performance in alkaline conditions.Micromodel experiments: tests were performed to evaluate the effect of AP in micromodels with a permeability contrast in a preliminary stage.Phase behavior and interfacial tension (IFT) experiments: these supported an understanding of the emulsion volumes generated by the fluid–fluid interactions.Two-phase core floods: we evaluated recovery at the core scale using various slugs; experiments were performed in real rock sand packs for live/dead oil conditions.

## 3. Materials

Reservoir Data: We evaluated alkali–polymers in the 8 Torton Horizon (TH) reservoir of the Matzen field in Austria in the Schoenkirchen area. As a clastic reservoir, the 8 TH reservoir is characterized by permeabilities between 150 mD and up to several darcys. The average porosity is 28–30%, the average net sand thickness is 5 m, and the reservoir temperature is 49 °C.

Oil Data: Oil from the Schoenkirchen S-85 well was used for this work. The oil is described as moderately degraded. A summary of the main data is presented in Table 1. Rupprecht [15] reports additional information on the oil. The oil is characterized by 39% saturated compounds, 42% aromatic, 16% resins, and an asphaltene content of about 3%. The oil’s saponifiable acids are about 42 µmol/g and the TAN number is 2.14 mg KOH/g. Dead oil density was measured as 0.931 g/cm^3^ (20 °C) and 0.891 g/cm^3^ (49 °C).

Synthetic Brines: Softened water produced from the field is envisaged as the injected water [7,9]. A simplified reservoir brine was composed in g/L of 22.47 NaCl, 0.16 KCl, 0.63 MgCl_2_*6 H_2_O, and 0.94 CaCl_2_*2 H_2_O, here named 8 TH RSB (reservoir synthetic brine). Moreover, 8 TH RSB was softened to create a simplified injection water, resulting in a composition in g/L of 22.62 NaCl, 0.16 KCl, and 1.52 NaHCO_3_ (buffer capacity), hereafter, Soft. 8 TH RSB. The pH for the alkali solutions is about 10.5 for 7.5 g/L Na_2_CO_3_ and the water viscosity at 49 °C was 0.65 mPa.s.

Alkali and Polymer: Na_2_CO_3_ was investigated here, as it is available at lower costs than other alkali agents. In addition, Na_2_CO_3_ is buffered at a pH of 10.2 for the conditions of the 8 TH reservoir and, hence, does not lead to substantial quartz dissolution (e.g., [16]) and scaling in the production wells accordingly. The high-molecular-weight anionic polyacrylamide (HPAM), a KemSweep A-5265 was used as the polymer. Three concentrations (1400 ppm, 1800 ppm, and 2000 ppm) were used, and polymer solution viscosity and concentration vary depending on the approach (aged or non-aged samples).

Outcrop Cores: Berea sandstone cores, as well as real rock material, were used for the experimental evaluations. Berea sandstone is a well-sorted yellowish sandstone with approximately 87% quartz, 5% feldspar, and 2.6% clay (Table 2). Cores with similar mineral compositions were used for the evaluations. The average values for Berea cores used in the two-phase tests were a 2.96 cm diameter, 29 cm length, a porosity of 0.219, and a brine permeability of 180–220 mD. Outcrop core samples were investigated using Computed Tomography (CT) scans for inhomogeneities. According to the obtained X-ray Diffraction (XRD) data, the clay is a mix of 92% kaolinite, 7.5% chlorite, and 0.7% illite by mass. Pore walls are covered with feldspar or clay, and additional data can be found in the previous work of Scheurer et al. [17], where cores from the same block were used.

Reservoir Material: We performed core floods using sand packs made of real rock material that was crushed to a uniform sand pile. Material from the reservoir section was used (Table 2). Sand pack characteristics, such as porosity and permeability, for the specific cases are presented in a subsequent section.

## 4. Experimental Methods

Thermal Aging of Polymer Solutions: Polymer solutions for thermal aging studies were prepared in a glove box under nitrogen atmosphere to ensure an anaerobic environment reflecting field conditions. The method is described more in detail in [13,14]. The aging was conducted in various concentrations. A concentration of 2000 ppm KemSweep A-5265 solution in Soft. 8 TH RSB brine in the presence of alkali was used for studying the polymer long-term viscosity performance at various temperatures. In addition, 1850 ppm and 1400 ppm solutions were used for aging samples for core flood testing, representing polymer performance down in the reservoir away from the nearby wellbore area. The pH of the solutions was 10.0–10.2 throughout the aging experiments.

Static Adsorption: Crushed and sieved Berea rock material with a grain size from 125 µm to 250 µm was used for static adsorption tests. 50 g of rock was mixed with 50 g of 200 ppm polymer solution in a jar. Sets of three identical samples were stored at 49 °C for 48 h. Afterwards, the polymer solution was filtered through a 5 µm syringe filter to remove the sand, and the final polymer concentration was determined utilizing Size Exclusion Chromatography (SEC) measurements. In case of crushed Matzen rock, the grains were washed from oil by Soxhlet extraction until clean before being used for adsorption tests. A polymer concentration of 200 ppm was selected for measurement accuracy.

Phase Behavior and IFT Tests: Tests were performed using the modified version of the procedure adopted by [18,19]. For this study, tests were performed in 60 mL glass tubes. First, 30 mL of aqueous phase (brine/cEOR) was filled in a precise syringe pump, and 30 mL of dead oil was added on the top. After sealing the tube, formulations were agitated strongly up and down for fifteen minutes to ensure homogenous mixing. Note that the oil was used as dead oil (DO) or dead oil mixed with cyclohexane (DOC). Subsequently, tubes were placed on the racks with temperature control on for 23 days. During the experiments, high-quality images were taken at specific time intervals to observe changes in the micro emulsions, if produced. Changes in pH value over time were expected to provide additional information about formulation reactions between alkali and high TAN oil. Interfacial tension (IFT Upper Phase/Middle Phase) at the fluids’ interface was calculated using the volume ratio between the upper phase (UP) and middle phase (MP) using the approach adopted by Liu et al. [20] and Hincapie et al. [9]. Interfacial tension data were reported in [13,14]; here, we compare results.

Micromodel Generation and Setup: Within this work, we customized and build a dual-permeability micromodel to mimic the heterogeneity of the reservoir. Furthermore, this enables us to screen the displacement efficiency of chemicals in a heterogenous formation. The micromodel dimensions are 60 × 20 mm, with a permeability contrast of 1:5 between the two zones. Micromodel generation and characterization were done by the same principle previously reported by Hincapie et al. [9]. Figure 1 displays the micromodel geometry and properties accordingly.

All micromodel experiments were carried out in the same setup reported by Schumi et al. [7]. The setup has an automated schedule feature, which allows us to program the experimental sequence of events to carry out experiments, eliminating human error and ensuring hardware reproducibility.

Micromodel Flooding Experiments—Sequence of Events: Figure 2 shows the pore volume injected for each individual slug used in all experiments. Injection velocity was 1 ft/day Darcy velocity, which translates to 0.2 μL/min.

Elapsed time, differential pressure, temperature at 4 positions (top, bottom, left, right), and high-resolution images were recorded during each experiment. Table 3 provides a summary of the parameters used for the experiments. Computation of saturation profiles was performed with an internal software (InspIOR Vision Pro). The software also enables computation for different sections chosen by the user. In this work, we specifically distinguished between the high and low-permeability layers as regions of interest.

Core Flood Experiments in Outcrops: For cEOR slug screening, five core floods (Table 4) were performed in Berea analogue using the core flooding setup reported in previous work [7,9]. The measured petrophysical properties of Berea plugs were similar to the targeted reservoir area (shown in the results section). Standard routine core analysis (RCA) was adopted for the initialization of the core plugs. Brine permeability was measured using the Darcy approach at three injection rates, while porosity and pore volume estimation was performed using the Archimedes method. Fresh and dry core samples were loaded in the Hassler cell, with a radial confining pressure of 30 bar(g). To displace air/nitrogen from the system, CO_2_ was flushed for ten minutes against the system pressure (back pressure regulator) of 10 bar(g). Following, the core was saturated with the appropriate brine (8 TH RSB (g/L)), with a system pressure of 5 bar(g). After measuring the permeability at room temperature, the system (oven) temperature was increased to the target temperature of 49 °C to validate permeability at reservoir temperature. After lowering the system temperature, the core plug was unloaded for porosity and pore volume estimation. Initial oil/brine saturations were achieved by dead oil flow through injection at a system pressure of 5 bar(g). The Dean–Stark procedure was applied to the collected fluids (oil/brine) to validate the visual volumes (saturations). Oil permeability was measured at three appropriate injection rates and the core sample was stored at target temperature for 4 weeks to age the samples and induce wettability alterations.

Core Flood Experiments in Real Rock—Sand Pack Preparation: Three sand pack experiments were performed using real rock material: two with dead oil and one with live oil (Table 5). The workflow for sand packs included cleaning the sand material, screening, matching the petrophysical properties to the target reservoir, and initial saturation of fluids. Unconsolidated material was crushed to a small particle size and was packed in Soxhlet extraction apparatus. Sand cleaning was performed by cooking an 80/20 mixture of chloroform/methanol at 60 °C for four days in repeated cycles until solvents were colorless. After cleaning, sand was dried in an oven at 60 °C for two weeks, then sieved through different mesh sizes to classify different grain sizes. The mixtures of two mesh sizes of 100–200 and 200–300 were mixed with the 87.5 wt% and 12.5 wt% to achieve the target permeability. Selected sand was loaded and packed into the Viton tube using a vibrator table until complete compaction was achieved. The final length of each sand pack was 30 cm, and they had a diameter of 3 cm. Further initialization and permeability calculations were similar to that of the core flood. However, porosity/PV estimation was performed by means of Nuclear Magnetic Resonance (NMR). The sand pack was fully saturated with methanol and was displaced by the formation brine (8 TH RSB). The brine/methanol mixture was analyzed by NMR to estimate the produced methanol volume. Further procedures of oil initialization, oil permeability, and the aging process were similar to those described in the previous section on core flooding. The used setup and further details on the approaches were also presented in [9].

Live Oil Preparation: One of the sand packs was re-saturated with live crude oil after the completion of the aging process with dead crude oil. Live oil preparation was performed in a high-pressure/temperature piston accumulator under reservoir conditions of 49 °C and a system pressure of 115 bar(g), adopting the procedure described in a previous study [9]. First, 150 mL of dead oil (S-85) was added to the piston accumulator; after sealing the piston, high-purity (4.5) methane gas was refilled in the piston to the target pressure of 115 bar (g). Dissolution of methane gas in dead oil was initiated by rotating the piston up and down. Gas dissolution resulted in lowering the system pressure, and the required pressure support was provided by an ICSO pump connected to the piston with a contact pressure mode. Equilibration and dissolution were established over a period of three days. In the end, the excessive amount of gas was bled off with help of a back pressure valve pre-set at 116 bar (g).

Core Flooding Experiments—Sequence of Events: Multiple fluid slugs were injected at an interstitial velocity of 1 ft/day, as shown in Figure 3. First, brine (8 TH RSB) was injected as slug 1 for all experiments for a 1.6 pore volume (PV) as a secondary mode. Second, polymer flooding was implemented as slug 2 (except Exp. 4 and Exp. 5) and was defined as a tertiary mode. Third, the alkali–polymer chemical slug was injected as a post tertiary mode (except Exp. 4 and Exp. 5) for 2 PV. Lastly, brine injection was implemented for 2 PV as a final slug for all experiments (diluted in synthetic brine 8 TH RSB). Core plugs/sand packs were unloaded after the flooding experiments to perform the Dean–Stark procedure. Final saturations (recovery factors) of the fractions’ collector samples were validated with Dean–Stark volumes. Core flood effluents from core plugs/sand packs were collected in small glass fraction tubes to perform volumetrics and to generate the produced oil recovery curve versus injected PV. Moreover, pressure differential data were recorded for all injected slugs.

## 5. Results

Thermal Aging: Long-term aging in the alkaline conditions was studied to find out the long-term polymer performance in AP applications. A rapid increase in the viscosity of the 2000 ppm solutions was observed during the first days at elevated temperatures (Figure 4A). The increase was linked to a fast change in the degree of hydrolysis providing a positive impact on viscosity in the soft brine. A more detailed description of the studies can be found in [13,14]. Even though the field temperature in Matzen is 49 °C, the aging was also carried out at 60 °C and 70 °C to accelerate the aging. We showed in [13,14] that the activation energy E_a_ = 110 kJ/mol, determined earlier for HPAM hydrolysis in neutral pH 6–8. Nurmi et al. [21] also applied it in the case of a high pH. With the knowledge of the activation energy, the aging results measured at one temperature can be transferred to the prediction at another temperature with the method described by Nurmi et al. [21]. Based on the activation energy, or E_a_ = 110 kJ/mol, the viscosity retention results in Figure 4A have been transferred to a corresponding time scale at 49 °C in Figure 4B.

The results suggest that the polymer gains viscosity up to 170% of the original viscosity in the present conditions within 1–2 months’ time. This indicates that the concentration for the injection could be lowered from the original planned concentrations to achieve the viscosity target. The viscosity of the polymer solution increases and reaches the target viscosity in the reservoir away from the wellbore. In order to compare the performance of the polymer after aging to a fresh polymer, samples at two different concentrations, 1850 ppm and 1400 ppm, were aged. The polymers were aged for 7 days at 70 °C in fully anaerobic conditions before using them in core flood or adsorption testing. These 7 days at 70 °C correspond to approximately 90 days at 49 °C. At this point, the viscosity has reached a level at which the rate of change is already very low [13,14]. The sampling point (aged samples) is indicated with black arrows in Figure 4A,B. The initial viscosity and viscosity after aging are shown in Table 6 for all the studied concentrations. A similar increase in viscosity, ~170%, was observed irrespective of the studied concentration. The target polymer viscosity in the application was set to 20 mPas. As seen in Table 6, this viscosity is reached with 1850 ppm of fresh polymer and with 1400 ppm of polymer that has aged in the AP conditions. Our results show that the acrylic acid content (degree of hydrolysis) increases rapidly. In contrast to observations at pH 6–8 (Nurmi et al. 2018), the hydrolysis rates are clearly not constant, but instead hydrolysis rates decrease over time. The decreasing polyacrylamide hydrolysis rate is explained by auto retardation: the hydrolyzing reagent at a high pH is an anionic OH- ion which is increasingly repelled by the anionic polymer as the charge content in the polymer chain increases.

Static Adsorption: Static adsorption experiments were carried to obtain insight into the adsorption phenomenon as a function of changing brine, pH, polymer charge, and rock material conditions. The results are presented in Figure 5. The trend in crushed Berea is clear. The adsorption decreases as the brine gets softer and the pH increases. Increasing the charge of the polymer in the softened brine conditions also appears to continue the positive trend. The lowest adsorption value measured was 19 µg/g. Adsorption in the actual cleaned Matzen rock material indicates even lower values. The adsorption in the 8 TH RSB brine was measured to be approximately half of that in the case of Berea: 27 µg/g vs. 48 µg/g, respectively. When applying the softer water with a pH increase in the Matzen rock, the adsorption could be expected to go even lower than with Berea, assuming a similar response to the change in conditions, as in the case of Berea. These values were not measured. Further considerations on the adsorption in the absence of oil can be found in [13,14].

Phase Behavior and IFT Tests: The data gathered include samples using dead oil (DO) or dead oil mixed with cyclohexane (DOC). From Figure 6A, a slight reduction in the *pH*-value of all formulations can be observed for the test period. This reduction in pH value was expected due to the chemical interactions between alkali and oil polar compounds. Over time, more alkali was neutralized with oil polar compounds and, hence, resulted in lowering the pH value. Considering the lower amount of alkali required and in order to hinder the possible alkali/polymer adsorption, a 7.500A-P(DOC) formulation was the optimum one and resulted in lowering the pH value. Figure 6B presents the reduction in produced emulsions over 23 days. The reduction in volume let us infer that the produced emulsions were instable. Moreover, the 7.500A-P(DOC) formulation produced a greater emulsion volume during the initial seven days. However, during the first two days, the 15.000A-P(DO) formulation produced the highest volumes of emulsions, but these were not stable and decreased the emulsion volume rapidly. Figure 7A,B presents the reduction in IFT for the chemical formulations. The observed decline in IFT was similar to that of the reduction in emulsion volume. The IFT calculation was performed using Chun Huh equation [20] (Table 7). Compared to our previous study [9], the IFT measured using the Chun Huh equation was in line with the spinning drop tensiometer. Hence, a lower amount of alkali in the 7.500A-P(DOC) formulation also depicted a good result in lowering the IFT to an order of 10^−2^. As can be seen from Figure 6, there was reproducibility in the lowering of the IFT in the experimental data for the phase behavior.

Micromodel Flooding Experiments: The micromodel experiments were performed stepwise to investigate the behavior of the fluid–fluid interactions. Without changing the injected fluid composition, we added additional slugs in each iteration of the experiments, as displayed in Figure 2.

In Table 8, the results are summarized and broken down into the obtained RF for each slug injection. Thus, we can quantify the efficiency of the injected chemical. Additionally, visual access through microfluidic technology gave us insights into the observed mechanisms during flooding. Most importantly, the impact of the cEOR fluids in the low-permeability layer and its desaturation behavior give a deeper understanding of recovery mechanisms. In the following section, the experiments are presented and discussed in detail.

In Exp. 1, the AP slug was injected in secondary mode to obtain a general idea about the fluid–fluid interactions between the AP and oil. The injection of the AP slug yielded an RF of 49% within the entire chip. This moderate desaturation performance is due to the remaining oil saturation (*S_or_*) in the low-permeability zone. Thus, the analysis of the results focused on both regions separately.

Figure 8a,b shows the saturation, pressure differential, and RF curves for Exp. 1. Differential pressure increases until a breakthrough is reached and stabilizes during further injection. Desaturation behaves according to the pressure response. Further injection recovers additional oil over time, and after 1 PV is injected, another drop in Sor is observed.

The post flush with brine showed no effect on oil displacement; yet, we observed an increase in differential pressure. From Figure 9 we observed a “light brownish” phase that was mobilized during the brine injection. The residual AP at the boundaries that remains within the porous structure further reacts with the oil; once the brine is injected, it manages to mobilize the oil at the boundaries. This lets us infer that the post flush displacement is not as effective as it is for a pure AP solution, thus leading to a brownish-colored phase.

From the visual access, we gained insights into the displacement behavior and efficiency of the AP at different permeability layer boundaries. In Figure 10, we have selected images at different stages of injection. The images on the left show the micromodel as it was photographed during the experiment and the images are processed with highlighted areas of displacement. We see that the AP has an RF of 87% from the high-permeability region, while the RF in the low-permeability region is only 14%. The efficient recovery from the lower layer is due to the AP, which favors IFT reduction and improvement of the mobility ratio. Recovery from the low-permeability layer happens in a slow and continuous manner. Low-permeability areas that are in contact with the AP are the ones where *S_or_* is reduced, namely, at the injection site and where the two layers meet in the middle.

For Exp. 2 (Figure 11) we introduced a brine injection prior to the AP. The first brine injection yielded an RF of 21%, followed by the AP slug, which recovered an additional 25%. The pressure differential increased almost twofold when injecting the AP; yet, after all the oil was mobilized, injectivity improved and reached values even below the previous brine injection. The final post brine flush did not recover barely any additional oil (1%), and the pressure dropped slightly further. Looking at the recovery from the two layers separately, the scenario in Exp. 2 shows a slightly worse performance compared to Exp. 1. In the high-permeability layer, the AP reached an additional RF of 45% after brine, while 5% was recovered from the low-permeability layer by the AP. Overall, the injection of the AP in secondary mode has been shown to be more effective in terms of the RF, and the “wash out” from the low-permeability zone at the layer boundaries was more efficient.

For comparison, in Exp. 3 (Figure 12) we targeted the experimental procedure as in Exp. 1, Exp. 2, and Exp. 3 from the core flooding section. The initial brine injection recovered 25%, followed by the polymer, which yielded only a 5% higher RF. The subsequent AP injection showed an additional 23% in RF and almost none in the post brine flush.

The pressure differential response showed an expected increase during the polymer injection. Once the oil bank was moved, the pressure declined. As soon as the AP slug entered the model, the pressure differential increased due to the mobilization of the oil; a decrease was then observed right away, thus improving injectivity. The final post brine flush slightly reduced the pressure, and the remaining AP within the pores was washed out by brine, which added a 1% increase in RF.

When looking at the layers separately, brine recovered 42% from the high-permeability layer and 1% from the lower-permeability layer. The polymer yielded an 11% RF from the higher layer and 0% from the lower-permeability layer. Only when the AP is injected, the RF from the lower-permeability layer is improved by 9%, while 32% is additionally recovered from the high-permeability zone. In summary (Figure 13), the overall RF was highest in Exp. 3 when all slugs were deployed, resulting in 54%. Secondary mode injection proved to be the most effective to displace oil from the low-permeability region and is second best in overall RF at 51%. Tertiary mode without a prior polymer injection yielded the lowest overall RF of 46%, with the least impact on recovery from the low-permeability layer.

Core Flood Experiments in Outcrops: Table 9 provides the summary of the flooding experiments performed in the Berea samples. Exp. 1 and Exp. 2 compared the role of the aging process for AP formulation as shown in Figure 14a,b. The oil recovery factors from brine flood in secondary mode were the same (≈3% difference). Interestingly enough, a similar trend of oil recovery (≈0.2% difference) from polymer flood was observed. The AP slug viscosity for Exp. 1 (38 mPa.s) is significantly higher than the one in Exp. 2, due to the thermal aging of the samples, which we previously reported in [13,14]. However, there was no significant oil recovery difference from the AP slugs in both experiments.

As expected, the pressure drop of the AP slug in Exp. 1 was higher than in Exp. 2, due to the formulation’s higher viscosity. This comparison let us infer that lowering the polymer concentration for thermally aged AP slugs is required to adjust the viscosity to 25 mPa.s. Lowering the polymer concentration from 1850 ppm to 1400 ppm for the thermally aged AP slug resulted in the required viscosity value. Contrary to the first two experiments, the brine flood and polymer flood resulted in lower oil recovery for Exp. 3, but the AP injection performed better, as shown in Figure 15. One possible reason could be that more oil saturation remained in Exp. 3 before the AP injection. Furthermore, slightly higher pressure drops were observed for all slugs in Exp. 3 compared to the first two experiments and can also be the result of higher residual oil saturation; this higher saturation of the oil resulted in the increase in pressure resistance at the inlet.

A comparison of Exp. 2 and Exp. 3 showed the potential in lowering the polymer concentration from 1850 ppm to 1400 ppm in the AP slug and, furthermore, to perform the slug aging process prior to injection to achieve the desired slug viscosity. Moreover, the comparison between Exp. 1 and Exp. 3 concludes that lowering the 450 ppm concentration reduces the oil recovery factor to only 2%, which falls within the marginal error of experimental repeatability. This reduction in the polymer concentration showed the potential to improve the utility factor (utility factor UF = kg of polymer injected/incremental bbl. oil produced by reducing the required number of polymers). Furthermore, lower pressure drops of the AP slug for Exp.3 were observed compared to Exp. 1, which could also improve the injectivity in field-scale applications.

Exp. 4 and Exp. 5 were only performed with an AP slug after a brine flood. The RF of the brine flood was in line with the previous experiments, but the AP formulation efficiency was better than the previous three experiments and resulted in 27% additional oil recovery. Exp. 4 and Exp. 5 concluded that, without a polymer slug prior to AP, the chemical interactions between the oil and the formulations were effective; hence, they resulted in better phase behavior interactions. Moreover, pressure drops for both experiments (4 and 5), as shown in Figure 14 were much lower than the previous experiments due to the absence of the pre-polymer slug. The oil recovery factor/reduction in oil saturation and pressure drop comparison between Exp. 4 and Exp. 5 confirmed that a good utility factor can be achieved by the aging process of the low-concentration AP slug.

Core Flood Experiments in Real Rock: Selected formulations from core flood experiments were further injected in reservoir sand packs, as shown in Figure 16 and Table 10. Contrary to the core flood experiments, the difference for the brine flood recovery factors of two sand packs (Exp. 6 and Exp. 7) was higher. The brine flood in the first two sand packs resulted in an almost 10% difference in oil recovery; however, the pressure drop data for both experiments were in line. One possible reason could be the different porosity of both sand packs. For screening purposes, the target was to match the permeability of the sand packs to the target reservoir permeability (300 mD). Similarly, regarding the AP chemical slugs of both experiments with almost the same bulk viscosity (≈24 mPa.s), the aged slug had a significantly lower oil recovery but resulted in a significantly higher pressure drop.

Based on the promising oil recovery results, the chemical formulations of Exp. 6 were injected into the sand pack using live oil. The injection was performed under reservoir conditions presented as Exp. 8 in Figure 16 and Figure 17. The RF from BF with live oil was higher compared to Exp. 7 but closer to Exp. 6. However, the polymer slug did not contribute any oil and the AP slug contributed to an additional 8% oil recovery. The additional oil contribution from the AP slug was similar to the thermally aged AP slug in Exp. 7. Hence, a comparison of the AP slugs between Exp. 7 and Exp. 8 let us conclude that the thermal aging could help to reduce the polymer amount required and contribute to additional oil recovery.

Economic Efficiency: To evaluate the economic efficiency of the various chemical agents, utility factors (UF = kg chemicals injected/incremental oil production) can be calculated for the core flood experiments. Here, we assume the injection of 1 PV of alkali–polymer, and the incremental oil recovery of the alkali–polymer slug observed. Note that we injected 1.5 PV in the laboratory to be sure that the displacement by the AP is complete. However, even considering dispersion, it is not necessary to inject more than 1 PV. That is why we used 1 PV for the one-dimensional EqUF calculation. Including the costs of the chemical agents, an Equivalent Utility Factor (EqUF) can be calculated:(1)$$EqUF=\frac{\frac{m_{P}*P_{P}+m_{C}*P_{C}+m_{A}*P_{A}+\ldots \ldots }{P_{P}}}{N_{Pinc}}\;\left[\frac{kg}{bbl}\right]$$
where *m* is the injected mass in kg of the individual components, P is the price of a component in USD/kg, and *NP_inc_* is the incremental oil recovery in bbl. Subscripts are P for polymer, C for co-solvent, and A for alkaline. If more components are utilized—e.g., a surfactant—the equation should be extended to n components. Here, we assume an Na_2_CO_3_ cost of 0.24 USD/kg and a polymer cost of 2.5 USD/kg. For the sake of simplicity, we focus here on comparing the experiments performed in outcrops.

The EqUFs for the injection of the aged and non-aged polymers with alkali for experiments 1–5 are shown in Figure 18. The EqUFs can be used to compare different chemical agents. Figure 18 shows that higher EqUFs are observed for experiments in which the aging of the polymer in an alkali solution in the reservoir was not accounted for. Reducing the polymer concentration in Experiment 3 (Figure 18) leads to a decrease in the EqUF by 40%. If the alkali–polymer is injected immediately without a prior polymer slug, then the economics are improved by 37% compared with the case (comparing Experiment 5 to Experiment 3). This is a good indication that it is beneficial to inject the alkali–polymer immediately rather than to start chemical EOR with a polymer injection. In the field test planned for the 8 TH reservoir, an area has been selected with an ongoing polymer flood for operational reasons, the roll out will be in an area of the field which is operated using waterflooding. The UFs of the polymer flood prior to the alkali–polymer are high, ranging from 9 kg/bbl. to 16 kg/bbl. for Experiments 1 to 3. The reason is the limited incremental oil observed in the experiments for polymer flooding. This shows the potential of the alkali–polymer EOR for reactive oils.

## 6. Discussion

Alkali–polymer injection is a promising technology to increase oil recovery in fields containing reactive oils. The initially generated soaps of the alkali with the reactive components of the oil can be measured to determine the amount of soap generated as a function of pH and the minimum required alkali concentration [22]. IFT measurements and phase behavior tests can be used to evaluate the presence of stable Windsor III emulsions (e.g., [23,24]) or thermodynamically instable emulsions (e.g., [9,25,26]) and the presence of salt–crude oil complexes at high alkali concentrations which are detrimental for enhanced oil recovery [27]. For the conditions of the 8 TH reservoir, a sufficient initial soap generation and the presence of instable emulsions were confirmed.

Two-layer micromodel experiments were used to investigate the displacement efficiency in heterogeneous rock. Micromodels have the advantage over sand packs or cores with two permeabilities because the boundary of the two permeability layers can be manufactured such that it neither creates a flow baffle nor a flow path. The results show that the highest incremental oil recovery over waterflooding is achieved from the high-permeability layer. Only at later stages does more crossflow occur, leading to increased recovery by AP flooding from the lower-permeability part of the micromodel. The results reveal that, to further increase recovery, polymer selection has to take the polymer molecular weight distribution and permeability distribution in the reservoir into account.

To improve the economics of AP flooding projects, the interaction of alkali with polymers over time needs to be considered. Aging polymers in alkali solution reveals the stability of the polymers but also the potential increase in polymer solution viscosity. For the conditions investigated here, the polymers were stable, and the viscosity increased with time. Displacement experiments showed that the recovery efficiency of the higher-concentration non-aged alkali–polymers is similar to the recovery efficiency of the lower-concentration but aged polymers. Hence, the polymer concentration in this AP project can be reduced compared with experiments using non-aged polymers. The polymer concentration is linked with the fracture half-length of the injection-induced fractures. The induced fractures might lead to the short circuiting of the injected fluids (e.g., [28]) and need to be included in the risk assessment for caprock integrity. Away from the wellbore, the viscosity of the polymers in alkali solution increases, leading to good sweep efficiency. Davidescu et al. [29] injected consecutive tracers and showed that the travel times for injected fluids decrease after polymer injection compared with water injection. The travel times are much longer than the required aging to increase the viscosity of the polymers in alkali. In the near wellbore, high flow velocities are observed and polymers in alkali solution are not aged and exhibit lower viscosities (neglecting visco-elastic effects) [30]. Further away from the wells, lower flow velocities (smaller arrows) are observed, and the polymer viscosity is increased. As shown in the section above, the EqUF is reduced by 40% by applying a lower polymer concentration. Furthermore, Nurmi et al. [14] showed that the adsorption of polymers on the rock is reduced in alkali solution compared to the case without alkali solution. Hence, the consumption of polymers in the reservoir is reduced, improving the economics of such projects.

## 7. Summary and Conclusions

Evaluating the synergies of alkali and polymers provided us with very important findings. On the one hand, polymer solutions need to be aged in alkali for alkali–polymer projects. On the other hand, the economics are much better if the alkali–polymer is injected directly. In the case studied here, we need and are planning to do the pilot in the polymer area for operational reasons.

We have shown that the performance of AP floods can be optimized by making use of lower polymer viscosities during injection but increasing polymer viscosities in the reservoir owing to the “thermal aging” of the polymers at a high pH. Furthermore, the AP conditions enable us to reduce the polymer retention in the reservoir, decreasing the utility factors (kg polymers injected/incremental bbl. produced).

The IFT measurements showed that the saponification at the oil–alkali solution interface very effectively reduces the IFT. Alkali phase experiments confirmed that emulsions are formed initially and supported the potential for residual oil mobilization. The aging experiments revealed that the polymer hydrolysis rate is substantially increased at a high pH compared to polymer hydrolysis at a neutral pH, resulting in a 60% viscosity increase in AP conditions. Micromodel experiments in two-layer chips depicted insights into the displacement. The two-phase experiments confirmed that lower-concentration polymer solutions aged in alkali show the same displacement efficiency as non-aged polymers with higher concentrations. Hence, significant cost savings can be realized, capitalizing on the fast aging in the reservoir. Due to the low polymer retention in AP floods, fewer polymers are consumed than in conventional polymer floods, significantly decreasing the utility factor (injected polymers kg/incremental bbl. produced).

We have shown that alkali/polymer (AP) injection leads to a substantial incremental oil production of reactive oils. A workflow was presented to optimize AP projects including reservoir effects. AP flood displacement efficiency must be evaluated incorporating the aging of polymer solutions. Significant cost savings and increasing efficiency can be realized in AP floods by incorporating the aging of polymers and taking the reduced polymer adsorption into account.

## Figures and Tables

**Figure 1 polymers-14-05508-f001:**
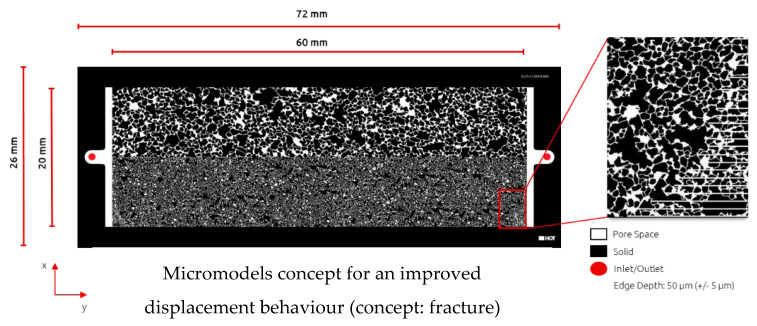
Illustration of the heterogenous micromodel used in this work. Top: high-permeability zone; Bottom: low-permeability zone.

**Figure 2 polymers-14-05508-f002:**
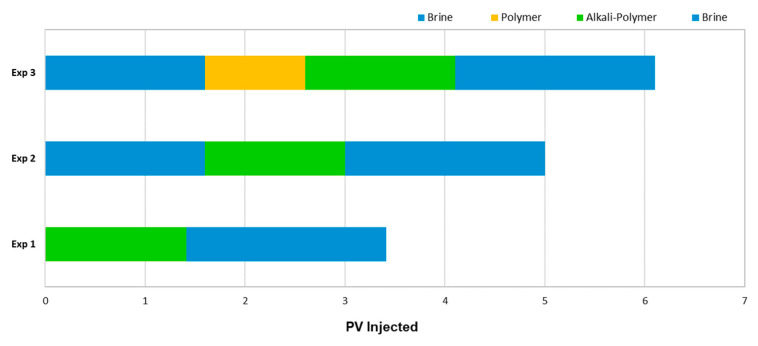
Summary of the injection sequences utilized for the different micromodel experiments.

**Figure 3 polymers-14-05508-f003:**
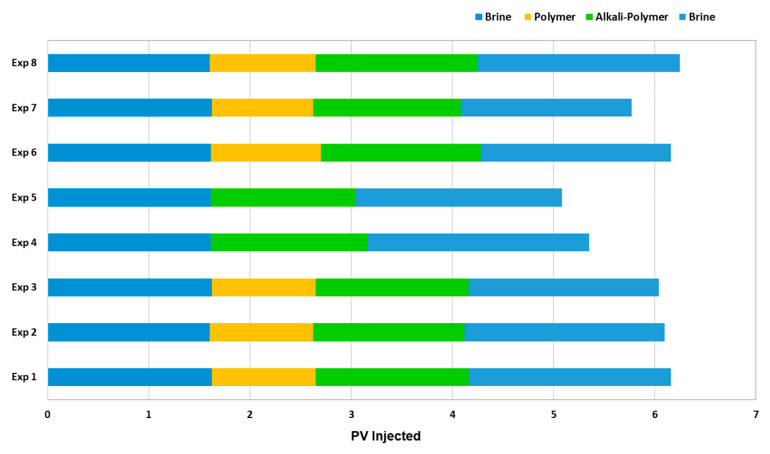
Injection sequences for the different core flood experiments utilized in this study.

**Figure 4 polymers-14-05508-f004:**
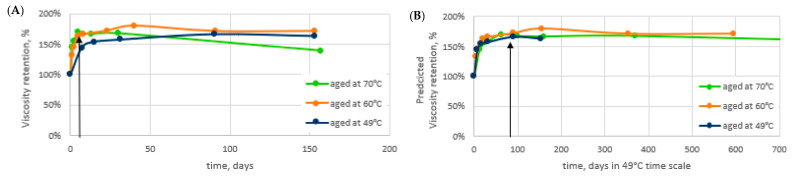
Aging results. Anaerobic 2000 ppm KemSweep A-5265 solutions in Soft. 8 TH RSB brine in the presence of alkali. (**A**) Aging in temperatures of 49 °C (dark blue), 60 °C (orange), and 70 °C (green). (**B**) aging results transferred to corresponding days at 49 °C. Black arrows indicate the point in time for the sampling of aged samples used in the core flood and adsorption testing.

**Figure 5 polymers-14-05508-f005:**
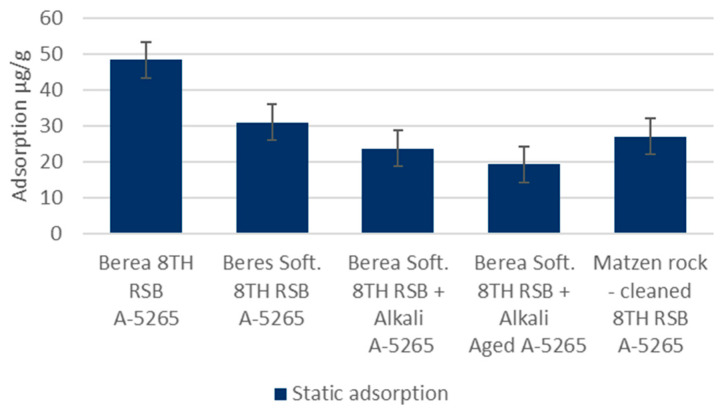
KemSweep A-5265 static adsorption in different conditions in crushed Berea and cleaned/crushed Matzen rock.

**Figure 6 polymers-14-05508-f006:**
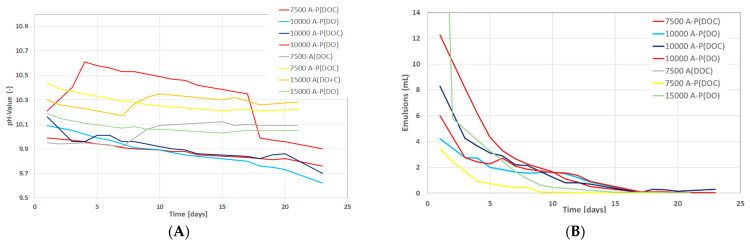
Change in pH value and emulsion volume over time for the phase behavior tests at different concentrations of chemicals. (**A**) Change in chemical formulation over number of days. (**B**) Change in produced emulsion volume over number of days. A refers to alkali concentration in ppm, P stands for polymer concentration of 1850 ppm, (DOC) represents dead oil with cyclohexane, and (DO) represents dead oil.

**Figure 7 polymers-14-05508-f007:**
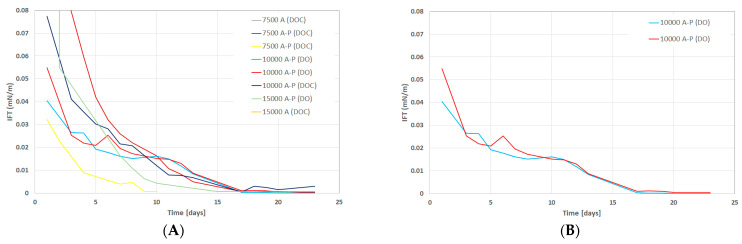
IFT variation over time (days) for the tested formulations in the phase behavior tests. (**A**) IFT versus time for various concentrations and (**B**) IFT versus time for 10,000 AP to show the data reproducibility. A refers to alkali concentration in ppm, P stands for polymer concentration of 1850 ppm, (DOC) represents dead oil with cyclohexane, and (DO) represents dead oil.

**Figure 8 polymers-14-05508-f008:**
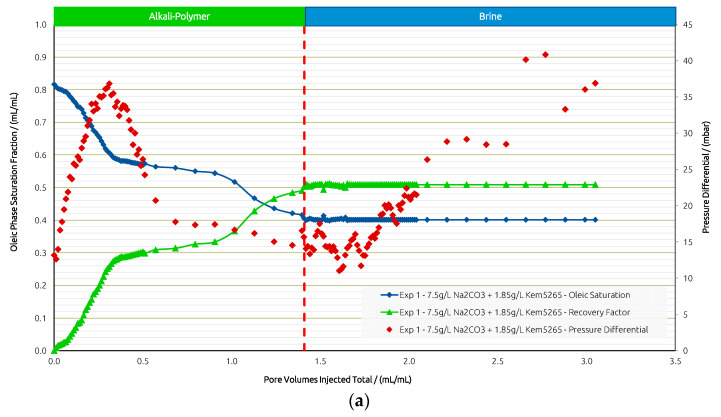
Summary of results obtained for Exp. 1 in micromodel experiments. (**a**) Oleic saturation, Recovery Factor (RF), and pressure differential for the entire chip and (**b**) RF of low-permeability zone vs. RF of high-permeability zone.

**Figure 9 polymers-14-05508-f009:**
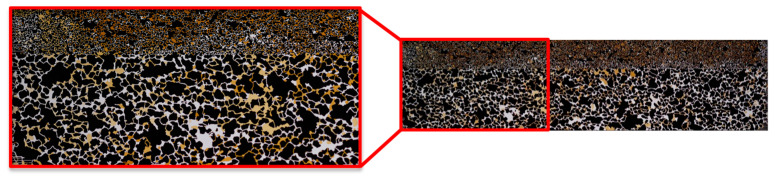
During brine injection, the remaining AP inside the porous structure reacted with the oil and the brine was displaced by an emulsion (light brown), resulting in an increase in pressure.

**Figure 10 polymers-14-05508-f010:**
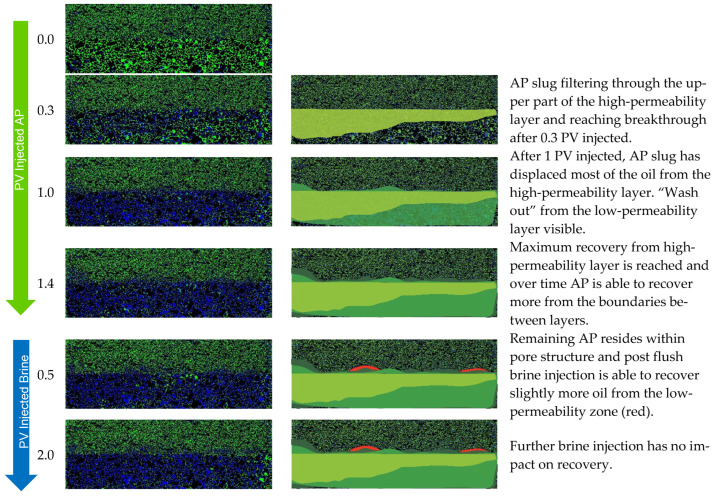
Displacement fronts during each slug injection in Exp. 1 (left: segmented images after analysis; right: post-edit images highlighting displaced area).

**Figure 11 polymers-14-05508-f011:**
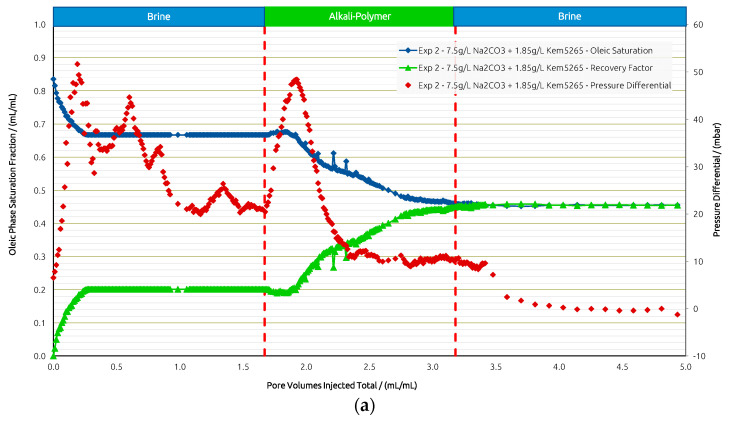
Summary of results obtained for Exp. 2 in micromodel experiments. (**a**) Oleic saturation, Recovery Factor (RF), and pressure differential for the entire chip and (**b**) RF of low-permeability zone vs. RF of high-permeability zone.

**Figure 12 polymers-14-05508-f012:**
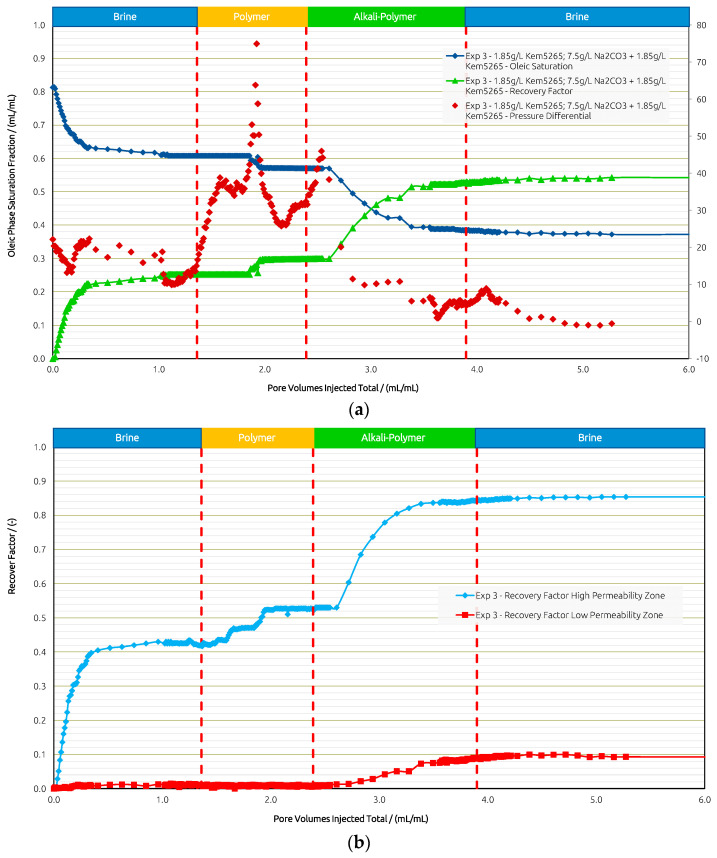
Summary of results obtained for Exp. 3 in micromodel experiments. (**a**) Oleic saturation, Recovery Factor (RF), and pressure differential for the entire chip and (**b**) RF of low-permeability zone vs. RF of high-permeability zone.

**Figure 13 polymers-14-05508-f013:**
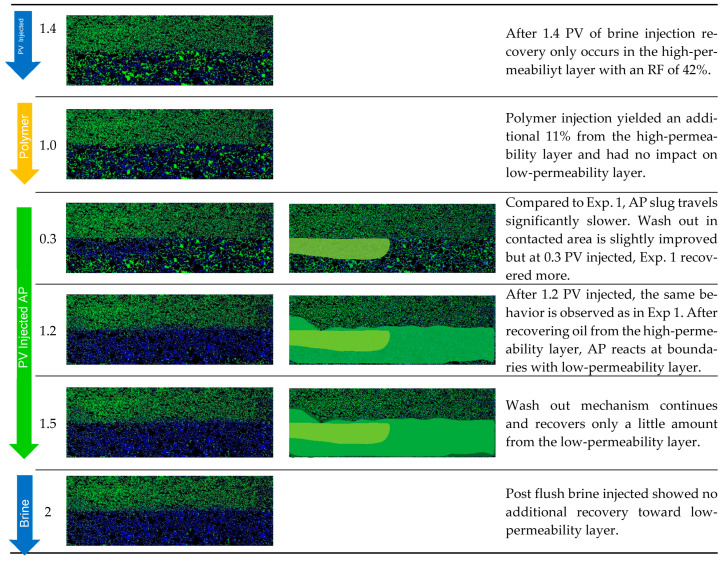
Displacement fronts during each slug injection in Exp. 3 (left: segmented images after analysis; right: post-edit images highlighting displaced area).

**Figure 14 polymers-14-05508-f014:**
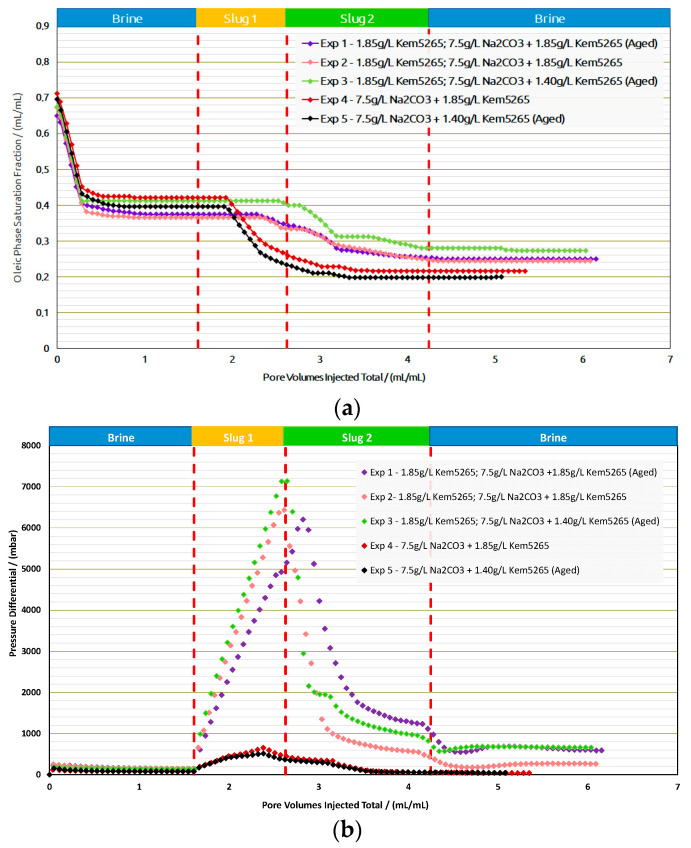
Oleic-phase saturation fraction/Pressure differential versus pore volumes injected total for the different slugs for core floods in Berea outcrops. (**a**) Reduction in oleic-phase saturation versus PV of slugs injected in Berea outcrops with crude dead oil initialization. (**b**) Pressure variation versus PV of injected slugs in Berea outcrops with dead oil initialization.

**Figure 15 polymers-14-05508-f015:**
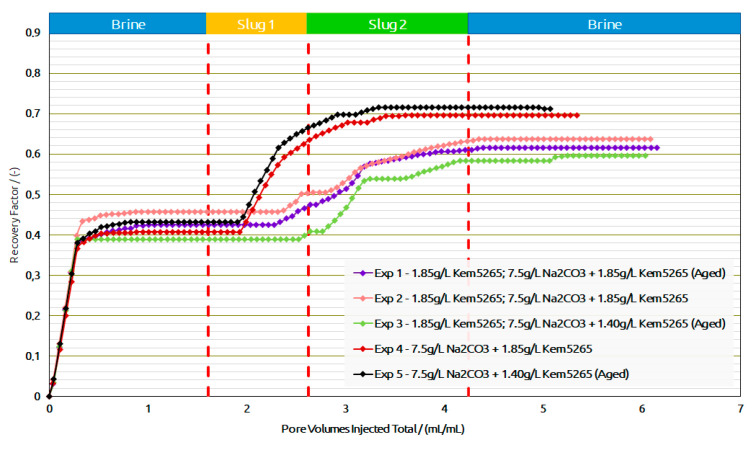
Recovery factor versus pore volumes injected total for the different slugs for core floods in Berea outcrops.

**Figure 16 polymers-14-05508-f016:**
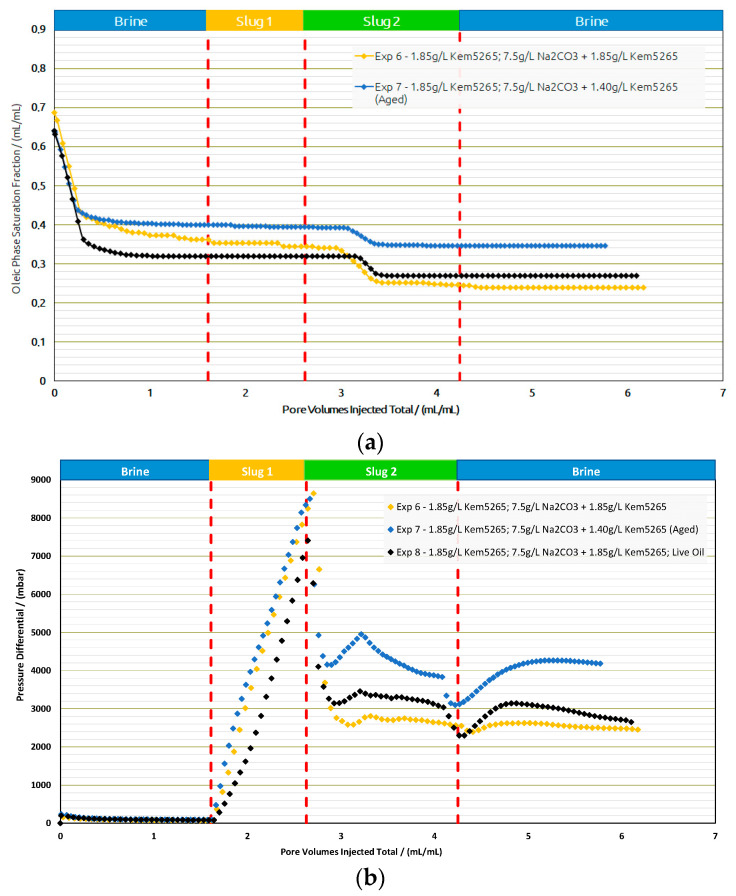
Oleic-phase saturation fraction/Pressure differential versus pore volumes injected total for the different slugs for core floods in real rock sand packs. Exp. 6 and 7 performed with dead oil and Exp. 8 with live oil. (**a**) Reduction in oleic-phase saturation versus PV of slugs injected in Berea outcrops with crude dead oil initialization. (**b**) Pressure variation versus PV of injected slugs in Berea outcrops with dead oil initialization.

**Figure 17 polymers-14-05508-f017:**
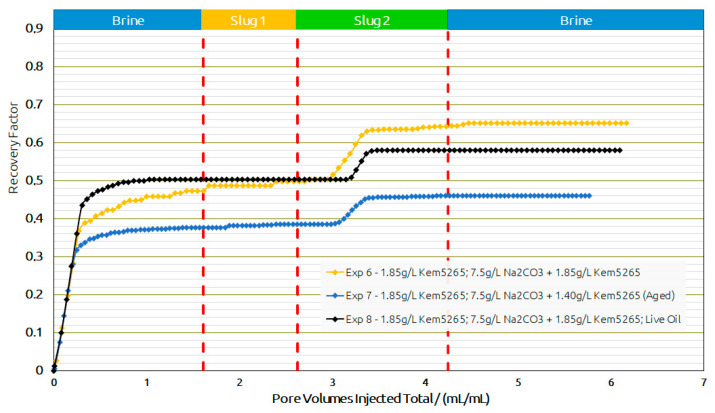
Recovery factor versus pore volumes injected total for the different slugs for core floods in real rock sand packs. Exp. 6 and 7 performed with dead oil and Exp. 8 with live oil.

**Figure 18 polymers-14-05508-f018:**
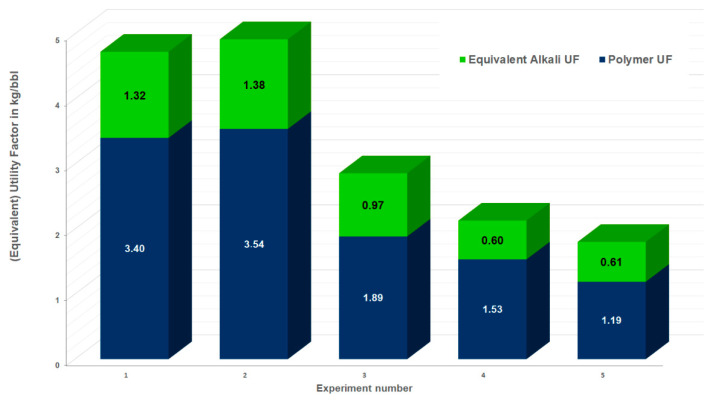
Equivalent utility factor (EqUF) for the experiments. The EqUF shown here reflects the incremental oil produced by the alkali–polymer slug and the mass of chemicals injected during this slug, assuming 1 PV injection. The polymer EqUF is the part of the overall EqUF related to the mass of the polymer injected during the injection of the AP slug, whereas the alkali EqUF reflects the cost-normalized mass of the alkali injected during the AP slug injection.

**Table 1 polymers-14-05508-t001:** Composition of crude oils used in this work from the 8 TH reservoir.

Property	Value
Well	Schoenkirchen S-85
TAN [mg KOH/g]	2.14
Saturates [%]	39
Aromatics [%]	42
Resins [%]	16
Asphaltene [%]	3
Saponifiable Acids [µmol/g]	41 (7.5 g/L Na_2_CO_3_)
µ (dead oil) @ 49 °C [mPa.s]	56
µ (live oil) @ 49 °C, 150 bar [mPa.s]	19
ρ (dead oil) @ 20 °C/49 °C [g/cm^3^]	0.931/0.891

**Table 2 polymers-14-05508-t002:** Mineral composition of the core material used in this work. Data reported in weight percent.

Core	Quartz	Kaolinite	K-Fsp	Calcite	Illite/Mica	Smectite	Peryte	Carbonate	Dolomite	Clay Tot
Berea Outcrop	89	-	7	0.8	-	-	0.3	-	-	2.9
Real Rock Reservoir	59.0	3	12	-	2.0	3.0	-	18	-	3

K-Fsp = Potassium feldspar.

**Table 3 polymers-14-05508-t003:** Summary of parameters in micromodel flood experiments performed in this work. Pore pressure was 1 bar and injection rate 1 ft/day in all cases.

Parameter	Unit	Experiment 1	Experiment 2	Experiment 3
Temperature	°C	49	49	49
P	-	-	-	1.85 g/L A−5265 *
Viscosity P	mPa.s	-	-	20
AP	-	7.5 g/L Na_2_CO_3_ + 1.85 g/L A-5265 **	7.5 g/L Na_2_CO_3_ + 1.85 g/L A-5265 **	7.5 g/L Na_2_CO_3_ + 1.85 g/L A-5265 **
Viscosity AP	mPa.s	25	25	25

* Diluted in synthetic brine 8 TH RSB (g/L): 22.53 NaCl, 0.16 KCl, 0.94 CaCl_2_*2 H_2_O, and 0.72 MgCl_2_*6 H_2_O ** Diluted in synthetic brine Soft. 8 TH RSB (g/L): 22.62 NaCl, 0.16 KCl, 1.52 NaHCO_3._

**Table 4 polymers-14-05508-t004:** Summary of parameters in Berea outcrop flood experiments performed in this work. Core orientation was vertical, pore pressure was 5 bar, injection rate was 1 ft/day, and radial pressure was 30 bar in all cases.

Parameter	Unit	Exp. 1	Exp. 2	Exp. 3	Exp. 4	Exp. 5
Temperature	°C	49	49	49	49	49
Chem. Slug 1 *	-	1.85 g/L A-5265			-	
ƞ Slug 1 (7.94 ^s−1^)	mPa.s	20	32	20	-	-
Chem. Slug 2 **	-	7.5 g/L Na_2_CO_3_ + 1.85 g/L A-5265 ^(1)^	7.5 g/L Na_2_CO_3_ + 1.85 g/L A-5265 ^(2)^	7.5 g/L Na_2_CO_3_ + 1.4 g/L A-5265 ^(1)^	7.5 g/L Na_2_CO_3_ + 1.85 g/L A-5265 ^(2)^	7.5 g/L Na_2_CO_3_ + 1.40 g/L A-5265 ^(1)^
ƞ Slug 2 (7.94 ^s−1^)	mPa.s	38	25	24	25	24

* Diluted in synthetic brine 8 TH RSB (g/L): 22.53 NaCl, 0.16 KCl, 0.94 CaCl_2_*2 H_2_O, and 0.72 MgCl_2_*6 H_2_O ** Diluted in synthetic brine Soft. 8 TH RSB (g/L): 22.62 NaCl, 0.16 KCl, 1.52 NaHCO_3_
^(1)^ Solution prepared according to aging procedure. ^(2)^ Solution prepared without aging procedure.

**Table 5 polymers-14-05508-t005:** Summary of parameters for sand pack flood experiments performed in this work. Core orientation was vertical and injection rate was 1 ft/day.

Parameter	Unit	Exp. 6	Exp. 7	Exp. 8 (Live)
Temperature	°C	49	49	49
Pore Pressure	Bar	5	5	125
Radial Confining Pressure	bar	30	30	150
Chem. Slug 1 *	-	1.85 g/L A-5265		
ƞ Slug 1 (7.94 ^s−1^)	mPa.s	20	21	19
Chem. Slug 2 **	-	7.5 g/L Na_2_CO_3_ + 1.85 g/L A-5265 ^(2)^	7.5 g/L Na_2_CO_3_ + 1.4 g/L A-5265 ^(1)^	7.5 g/L Na_2_CO_3_ + 1.85 g/L A-5265 ^(2)^
ƞ Slug 2 (7.94 ^s−1^)	mPa.s	25	23	29

* Diluted in synthetic brine 8 TH RSB: 22.53 g/L NaCl, 0.16 g/L KCl, 0.94 g/L CaCl_2_*2 H_2_O, and 0.72 g/L MgCl_2_*6 H_2_O ** Diluted in synthetic brine Soft. 8 TH RSB: 22.62 g/L NaCl, 0.16 g/L KCl, 1.52 NaHCO_3._
^(1)^ Solution prepared according to aging procedure. ^(2)^ Solution prepared without aging procedure.

**Table 6 polymers-14-05508-t006:** Summary of concentrations and viscosities used here. Viscosity increases after aging, averaging 165%.

Polymer Concentration	Viscosity in Soft. 8 TH RSB + Alkali (mPas)	Viscosity after Aging in Soft. 8 TH RSB + Alkali (mPas)	Increase in Viscosity
2000	24.6	41.7	168%
1850	20.5	34.4	168%
1400	12.0	20.4	170%

**Table 7 polymers-14-05508-t007:** Summary of volume relation from phase behavior and IFT for 7500 ppm A and AP determined using the Chun Huh equation (Liu et al. 2008) for alkali and alkali–polymer in 10 mL oil sample. IFT = Ratio Vphase/Vphase * c, c = 0.3 nM/m (Liu et al. 2008)—correction factor. Lower phase = water, middle phase = emulsion, upper phase = oil.

	Time [Days]	Lower Phase (LP) Vol. [ml/mL]	Middle Phase (MP) Vol. [ml/mL]	Upper Phase (UP) Vol. [ml/mL]	Ratio $\frac{V_{Middle\;Phase}}{V_{Lower\;Phase}}$ [–]	IFT _MP/LP_ [mN/m]	Ratio $\frac{V_{Middle\;Phase}}{V_{Lower\;Phase}}$ [–]	IFT _UP/MP_ [mN/m]
A	7	0.000	0.9674	0.033	5.979	1.7940	1.4705	0.4412
	10	0.091	0.5413	0.368	1.203	0.3611	0.9742	0.2922
	16	0.331	0.3987	0.409	1.189	0.3567	0.9225	0.2768
	17	0.321	0.3816	0.414	1.166	0.3499	0.8879	0.2664
	18	0.317	0.3697	0.416	1.113	0.3339	0.7543	0.2263
	21	0.294	0.3272	0.434	5.979	1.7940	1.4705	0.4412
AP	5	0.4137	0.0721	0.5142	0.1742	0.0523	0.1401	0.0420
	8	0.4518	0.0375	0.5108	0.0829	0.0249	0.0734	0.0220
	12	0.4755	0.0141	0.5104	0.0297	0.0089	0.0277	0.0083
	18	0.4895	0.0013	0.5092	0.0026	0.0008	0.0025	0.0007
	20	0.4902	0.0013	0.5085	0.0026	0.0008	0.0025	0.0007
	23	0.4892	0.0003	0.5105	0.0006	0.0002	0.0006	0.0002

**Table 8 polymers-14-05508-t008:** Summary of results obtained for micromodel flood experiments performed in this work. Pore pressure was 1 bar and injection rate was 1 ft/day in all cases. Temperature, 49 °C. High-permeability zone refers to 6 darcys and low-permeability zone to 1.5 darcys.

Parameter	Units	Experiment 1	Experiment 2	Experiment 3
Polymer (P)	-	-	-	1.85 g/L A-5265 *
Ƞ Polymer (7.94 ^s−1^)	mPa.s	-	-	20
Alkali–Polymer (AP)	-	7.5 g/L Na_2_CO_3_ + 1.85 g/L A-5265 **	7.5 g/L Na_2_CO_3_ + 1.85 g/L A-5265 **	7.5 g/L Na_2_CO_3_ + 1.85 g/L A-5265 **
Ƞ AP (7.94 ^s−1^)	mPa.s	25	25	25
So initial	%	82	84	81
So final	%	40	46	37
RF (Brine)	%	not applicable	21	25
Add. RF (P)	%	not applicable	not applicable	5
Add. RF (AP)	%	49	25	23
Add. RF (Brine)	%	2	1	1

* Diluted in synthetic brine 8 TH RSB: 22.53 g/L NaCl, 0.16 g/L KCl, 0.94 g/L CaCl_2_*2 H_2_O, and 0.72 g/L MgCl_2_*6 H_2_O ** Diluted in synthetic brine Soft. 8 TH RSB: 22.62 g/L NaCl, 0.16 g/L KCl, 1.52 NaHCO_3_.

**Table 9 polymers-14-05508-t009:** Summary of results obtained for Bentheimer outcrop flood experiments performed in this work. Core orientation was vertical, pore pressure was 5 bar, injection rate was 1 ft/day, and radial pressure was 30 bar in all cases. Viscosities are taken at 7.94 s^−1^.

Parameter	Unit	Exp. 1	Exp. 2	Exp. 3	Exp. 4	Exp. 5
Temperature	°C	49	49	49	49	49
Slug 1	g/L	1.85 P *	1.85 P *	1.85 P *	-	-
ƞ Slug 1	mPa.s	20	20	20	-	-
Slug 2 *	g/L	7.5 A + 1.85 P (aged) ^(1)^	7.5 A + 1.85 P	7.5 A + 1.4 P (aged) ^(1)^	7.5 A + 1.85 P	7.5 A + 1.4 P (aged) ^(1)^
ƞ Slug 2	mPa.s	38	25	24	25	24
Length	cm	30.00	30.00	29.90	30	30.1
Diameter	cm	3.81	3.80	3.81	3.81	3.80
Bulk Volume	cm3	342.07	343.94	342.02	343.94	341.37
PV	cm3	75.10	77.54	75.97	71.67	70.38
Porosity	%	22.0	22.5	22.0	22.2	20.5
Perm. (*k_w_*)	mD	271	277	269	369	356
Oil Sat. Init.	%	65	67	67	0.71	0.70
RF brine	%	42.5	45.6	38.8	41	43
RF Polymer	%	4.9	4.7	2.0	-	
RFAP	%	13.3	12.4	17.6	27	26.7
RF Brine	%	0.9	1.0	1.1	2	1.4

* Diluted in synthetic brine 8 TH RSB: 22.53 g/L NaCl, 0.16 g/L KCl, 0.94 g/L CaCl_2_*2 H_2_O, and 0.72 g/L MgCl_2_*6 H_2_O. ^(1)^ Solution prepared according to aging procedure. P = A- 5265 A = Na_2_CO_3_ PV = Pore Volume.

**Table 10 polymers-14-05508-t010:** Summary of results obtained for sand pack flood experiments performed in this work. Core orientation was vertical and injection rate was 1 ft/day Viscosities are taken at 7.94 s^−1^. Experiments performed at 49 °C.

Parameter	Unit	Exp. 6	Exp. 7	Exp. 8 (Live)
Pore Pressure	Bar	5	5	125
Radial Pressure	bar	30	30	150
Slug 1	g/L	1.85 P *	1.85 P *	1.85 P *
ƞ Slug 1	mPa.s	20	21	20
Slug 2 *	g/L	7.5 A + 1.85 P	7.5 A + 1.4 P (aged) ^(1)^	7.5 A + 1.85 P
ƞ Slug 2	mPa.s	25	23	24
Length/Diameter	cm	29.60/2.99	30.35/2.99	30.55/2.99
Bulk Volume/PV	cm3	207.73/69.2	213.10/88.0	214.51/73
Porosity	%	33.3	41.3	34
Perm. (*k_w_*)	mD	297	230	286
Oil Sat. Init.	%	69	64	64
RF brine	%	47.3	37.6	50
RF Polymer	%	2.5	0.9	0
RFAP	%	14.6	7.4	8
RF Brine	%	0.8	0.1	0

* Diluted in synthetic brine 8 TH WTP: 22.53 g/L NaCl, 0.16 g/L KCl, 0.94 g/L CaCl_2_*2 H_2_O, and 0.72 g/L MgCl_2_*6 H_2_O. ^(1)^ Solution prepared according to thermal aging procedure. P = A − 5265; A = Na_2_CO_3_; PV = Pore Volume.

## Data Availability

Not applicable.

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
