# Peer review of "Optimizing Polymer Costs and Efficiency in Alkali–Polymer Oilfield Applications"

_polymers, 2022, doi:10.3390/polym14245508_

Round 1

Reviewer 1 Report

The manuscript systematically investigated the optimising cost and efficiency. I think the research is well-designed and the results are well-presented. The only suggestion that I have is to add some statistical data in the abstract so that the reader can gain an insight about the whole paper.

Therefore, I suggest the acceptance of the manuscript. 

Author Response

Dear Reviewer 1

Thank you for inviting us to submit a revised draft of our manuscript polymers 2067740. We also appreciate the time and effort you have dedicated to providing insightful feedback on ways to strengthen our paper. Thus, it is with great pleasure that we resubmit our article for further consideration.  

We have incorporated changes that reflect the detailed suggestions you have kindly provided. We also hope that our edits and the responses we provide below satisfactorily address all the issues and concerns you have noted.

The manuscript systematically investigated the optimising cost and efficiency. I think the research is well-designed and the results are well-presented. The only suggestion that I have is to add some statistical data in the abstract so that the reader can gain an insight about the whole paper.

Therefore, I suggest the acceptance of the manuscript. 

Many thanks for your comment and nice words. Indeed, we believe there is substantial value in making use of the polymer alkali interaction. Abstract is updated accordingly.

Reviewer 2 Report

The manuscript describes the usefulness of a polymer in oilfield applications. It is well written but it needs corrections at some places and complementary information.

There are some mistakes in writing, for example "de" instead of "the" in Abstract, 10-2 instead of 10-2 (page 10, line 369), 10-ML instead of 10 ml (caption of Table 7).

Is symbol "c" for concentration in caption of Figure 7? Check if dimension nM/m is correct in "c=0.3nM/m".

The following sentence is hard to understand (page 11, line 415): "Displacement is not as effective as for a pure AP solution, thus the brownish color."

A structural formula of the used alkaline polymer could improve the presentation of results.

The authors mentioned that AP further reacts with oil. What type of reactions can be expected and with what components?

What type of processes can take place in polymer aging (for example aggregation)?

I recommend for the better highlighting to put into "..." format the next sentence in page 21, line 679: "Error! Reference source not found."

Author Response

Dear Reviewer 2

Thank you for inviting us to submit a revised draft of our manuscript polymers 2067740. We also appreciate the time and effort you have dedicated to providing insightful feedback on ways to strengthen our paper. Thus, it is with great pleasure that we resubmit our article for further consideration.  

We have incorporated changes that reflect the detailed suggestions you have kindly provided. We also hope that our edits and the responses we provide below satisfactorily address all the issues and concerns you have noted.

The manuscript describes the usefulness of a polymer in oilfield applications. It is well written, but it needs corrections at some places and complementary information.

Many thanks for your comment and nice words. Indeed, we believe there is substantial value in making use of the polymer alkali interaction.

There are some mistakes in writing, for example "de" instead of "the" in Abstract, 10-2 instead of 10-2 (page 10, line 369), 10-ML instead of 10 ml (caption of Table 7).

Thank you for the note. We adapted the requested changes.

Is symbol "c" for concentration in caption of Figure 7? Check if dimension nM/m is correct in "c=0.3nM/m".

Thanks for your comment, c refers to the correction factor c=0.3 nM/m taken from the work of Liu et al. 2008

The following sentence is hard to understand (page 11, line 415): "Displacement is not as effective as for a pure AP solution, thus the brownish color."

Thanks for your comment, we have changed the sentence to: The residual AP at the boundaries that remains within the porous structure further reacts with the oil, once brine is injected it manages to mobilize oil at the boundaries. This let us infer that the post flush displacement is not as effective as for a pure AP solution, hence leading to a brownish colored phase.

A structural formula of the used alkaline polymer could improve the presentation of results.

Great comment, many thanks. Although we have deepened into the chemical reactions taking place, we see a bit risky taken that route without having the in deep insights to conclude.

The authors mentioned that AP further reacts with oil. What type of reactions can be expected and with what components?

Thanks for your comment, we refer to the reaction of the polar components (saponifiable acids) and the alkali molecules. Here, we were wondering how the complex interplay of increasing the viscosity (AP) and generation of soaps at the oil-alkali solution interface would lead to increased oil recovery.

What type of processes can take place in polymer aging (for example aggregation)?

Thanks for your comment. We have not fully considered aggregation taking place, since we did not analyze in detail the reaction staking place. Our results show that that acrylic acid content (degree of hydrolysis) increases rapidly. In contrast to observations at pH 6-8 (Nurmi et al. 2018), the hydrolysis rates are clearly not constant but instead hydrolysis rates decrease over time. Decreasing polyacrylamide hydrolysis rate is explained by auto retardation: the hydrolyzing reagent at high pH is anionic OH- ion which is increasingly repelled by the anionic polymer as the charge content in the polymer chain increases.

I recommend for the better highlighting to put into "..." format the next sentence in page 21, line 679: "Error! Reference source not found."

Thank you for the note. We adapted the requested changes.